Medical Imaging with Deep Learning – Accepted 2023                    Short Paper – MIDL 2023

# Nearest Neighbor Radiomics for Self-Supervised Chest X-ray Pneumonia Identification

**Cailin Winston**[*1,2]                                   CAILINW@CS.WASHINGTON.EDU

**Caleb Winston**[*1]                                      CALEBWIN@CS.STANFORD.EDU

**Chloe Winston**[2,3]                        CHLOE.WINSTON@PENNMEDICINE.UPENN.EDU

[1] *Department of Computer Science, University of Washington, Seattle (UW),* [2] *Department of Biochemistry, UW,* [3] *Department of Neuroscience, UW*

## Abstract

Self-supervised training minimizes a contrastive loss objective for unlabeled data. Contrastive loss estimates the distance in the latent space between positive pairs, which are pairs of images that are expected to have the same label. For medical images, choosing positive pairs is challenging because simple transformations like rotations or blurs are not class-invariant. In this paper, we show that choosing positive pairs with nearest-neighbor radiomics features for self-supervised training improves chest X-ray pneumonia identification accuracy by 8.4% without labeled data.

**Keywords:** Contrastive Learning, Self-Supervised Learning, Radiomics, Chest X-Ray, Pneumonia Identification

## 1. Introduction

Many diseases affecting the lungs, such as pneumonia, can be diagnosed by human analysis of chest X-rays. However, variability in radiologists' interpretations has motivated the use of deep learning models (Neuman et al., 2012) for automatic disease identification. Because these models require large amounts of labeled training data (Saraiva et al., 2019), methods such as transfer learning (Kundu et al., 2021) and contrastive learning (Han et al., 2021) are promising. For example, NNCLR maximizes the similarity between latent embeddings of positive pairs of images (nearest neighbors in embedding space) (Dwibedi et al., 2021). Although successful for natural image classification, NNCLR and other contrastive learning techniques do not directly extend to medical image classification, because visually similar or geometrically transformed medical images can have profoundly different pathology.

Thus, we propose an approach for self-supervised training in medical imaging that maximizes similarity in latent embedding space between different images that have nearest neighboring radiomics features. Radiomics reduces an image to a set of biologically meaningful and radiologist-interpretable features (Tomaszewski and Gillies, 2021). We hypothesize that self-supervised training with nearest-neighbor radiomics will learn latent embeddings that reflect variation in radiomics features, which better predict pathology. In this paper, we discuss our approach and evaluate it on chest X-ray pneumonia identification.

## 2. Methods

We propose using nearest-neighbor radiomics (NN-radiomics) to identify positive pairs for self-supervised training of chest X-ray classification models. The end-to-end methodology is (1) pretraining on a general labeled dataset such as ImageNet, (2) self-supervised pretraining with NN-radiomics and an algorithm for self-supervised learning such as SimSiam, and (3) supervised fine-tuning for the specific task such as pneumonia identification.

---

[*] Equal contribution

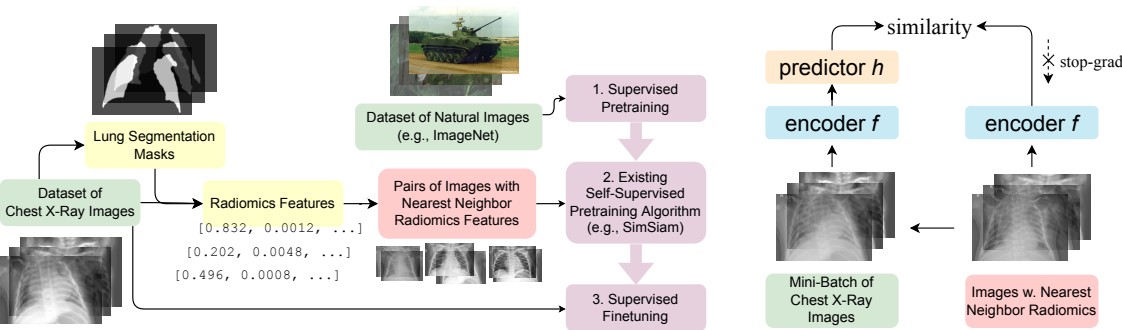

(a) End-to-End Methodology for Training Chest X-Ray Models with NN-Radiomics SSL.

(b) Architecture for SimSiam-based NN-radiomics SSL.

Figure 1: Nearest-Neighbor Radiomics (NN-Radiomics) Self-Supervised Learning (SSL)

## 2.1. Self-supervised Pretraining with Nearest-Neighbor Radiomics

We use an off-the-shelf image segmentation model to extract the lungs from each chest X-ray image (Selvan et al., 2020). Then, we compute standard radiomics features using PyRadiomics (Ferreira Junior, 2021). We used 94 our of the 120 available features (excluded "shape-based" features). For each unlabeled image, we then find its nearest neighboring image using the computed radiomics features. This produces a set of positive pairs with nearest neighboring radiomics. The model is then pretrained using any existing self-supervised algorithm, such as SimSiam (Chen and He, 2021) - which we used - or BYOL.

## 2.2. Model Architecture

The model architecture (Figure 1(b)) for chest X-ray classification consists of an encoder and a classifier (not pictured). The encoder is a backbone model (ResNet18 (Kaiming He and Sun, 2016) pretrained on ImageNet) with a projection MLP head and the classifier is a linear layer with an output size of 1. The parameters of the encoder are learned in the self-supervised pretraining step, and the classifier is trained on frozen features from the backbone component of the encoder during the supervised finetuning step.

## 2.3. Experimental Setup

We evaluated self-supervised training with NN-radiomics on a chest X-ray model for pneumonia detection. We used a binary pneumonia identification dataset of 5856 chest X-rays (Kermany, 2018) that we randomly split into pretraining, finetuning, testing, and validation splits in a 7:2:0.9:0.1 ratio. The model was pretrained for 20 epochs and finetuned for 60.

## 3. Results

**RQ1: Is NN-radiomics a high-quality indicator of task-specific positive pairs?**
We found that in a dataset of 4172 chest X-rays, 85.69% of nearest neighbor radiomics (NN-radiomics) positive pairs have the same label (pneumonia vs. control). Furthermore, 65.91% with pneumonia have the same type of pneumonia (viral vs. bacterial). The high

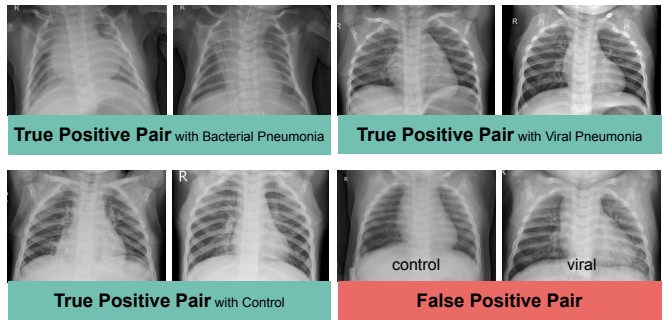

Figure 2: Examples of positive pairs with nearest neighboring radiomics features.

percentage of NN-radiomics positive pairs with same labels motivates learning similar latent embeddings via contrastive learning for NN-radiomics positive pairs.

Table 1: Accuracy on Pneumonia Identification

| Method | Accuracy (F1) | AUROC |
|---|---|---|
| Baseline | 0.7996 | 0.9313 |
| Pretraining w. Random Positive Pairs | 0.8141 | 0.9311 |
| Pretraining w. 1st-NN-Radiomics Positive Pairs | 0.8669 | 0.9517 |
| Pretraining w. 10th NN-Radiomics | 0.8557 | 0.9572 |
| Pretraining w. 50th NN-Radiomics | 0.8381 | 0.9537 |

**RQ2: Can self-supervised pretraining with NN-radiomics boost accuracy?** Our results in Table 1 demonstrate a notable boost in accuracy by pretraining with NN-radiomics-based contrastive learning compared to the baseline of no pretraining.

**RQ3: How does nearness of radiomics positive pairs affect accuracy?** We conduct an ablation on $n$ where the $n$th-nearest radiomics pairs are used for self-supervised learning. The results in Table 1 demonstrate that accuracy degrades as the positive pairs used are farther in radiomics space.

## 4. Conclusion

We present an approach to pretraining chest X-ray models without labeled data by using positive pairs that have nearest neighboring radiomics features. Our results demonstrate a notable improvement in pneumonia identification accuracy through self-supervised pretraining of chest X-ray models using nearest-neighbor radiomics.

## Acknowledgments

We thank Dr. Linda Shapiro, Professor of Computer Science and Engineering at the University of Washington for her support.

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
