# OpenReview forum: "Nearest Neighbor Radiomics for Self-Supervised Chest X-ray Pneumonia Identification"
_MIDL.io/2023/Short_Paper_Track — MIDL 2023 Short paper track Poster_

### Official Review · Reviewer_W4m8 · 2023-04-21
**Interesting approach for selecting image pairs**

**Rating:** 6
**Confidence:** 3

**Review:**

The paper proposes to select pairs for contrastive training, based on nearest neighbors defined on a standard set of radiomics features. The accuracy/AUC is higher for picking pairs based on this approach, than picking random pairs.

Pros:
* Nice idea for selecting pairs (though more details about the radiomics features are needed)
* Clear overview of the method

Cons:
* More details about the radiomics features needed, what is the dimensionality?
* Single data split / single performance estimates reported, cross-validation would have been more appropriate given the data size

---

### Official Review · Reviewer_XRx6 · 2023-04-24
**Interesting radomics-based contrastive learning**

**Rating:** 7
**Confidence:** 4

**Review:**

This paper investigates a self-supervised learning approach for Chest X-ray pneumonia identification. The method encourages high affinities between the embeddings of images having close radiomics features. It is well-known that radiomics reflect biologically meaningful and radiologist-interpretable features. Therefore, this study hypothesizes that self-supervised training with nearest-neighbor radiomics would learn better embeddings and yield better pathology predictions.

I concur with this hypothesis. Self-supervised learning with such information makes much more sense than using geometric augmentations (which are more appropriate for natural images). Experimentally, the authors show the value of the approach in a standard pre-train and fine-tune setting: (1) self-supervised pre-training based on SiamSiam (Chen and He, 2021) and radiomics; and (2) supervised fine-tuning for the specific task of pneumonia identification.

In my opinion, this is a good paper with interesting findings. Also, the approach is well motivated. For instance, the authors found that, in a dataset of 4172 chest X-rays, 85.69% of nearest neighbor radiomics pairs have the same label (pneumonia vs. control). These high percentages of radiomics-based positive pairs with the same labels motivate clearly supervising pairwise contrastive learning with radiomics information.